# Development of C-Shaped Parasitic MIMO Antennas for Mutual Coupling Reduction

Hamizan Yon [1,*](ORCID), Nurul Huda Abd Rahman [1,*](ORCID), Mohd Aziz Aris [2], Mohd Haizal Jamaluddin [3](ORCID), Irene Kong Cheh Lin [3], Hadi Jumaat [4], Fatimah Nur Mohd Redzwan [2] and Yoshihide Yamada [5]

1   Antenna Research Centre, School of Electrical Engineering, College of Engineering, Universiti Teknologi MARA, Shah Alam 40450, Selangor, Malaysia
2   School of Electrical Engineering, College of Engineering, Universiti Teknologi MARA, Cawangan Terengganu, Kampus Dungun, Dungun 23000, Terengganu, Malaysia; mohda474@uitm.edu.my (M.A.A.); fatima804@uitm.edu.my (F.N.M.R.)
3   Wireless Communication Centre (WCC), School of Electrical Engineering, Universiti Teknologi Malaysia, Skudai 81310, Johor, Malaysia; haizal@fke.utm.my (M.H.J.); irenekongchehlin@gmail.com (I.K.C.L.)
4   School of Electrical Engineering, College of Engineering, Universiti Teknologi MARA, Kota Samarahan 94300, Sarawak, Malaysia; hadi431@uitm.edu.my
5   Malaysia-Japan International Institute of Technology (MJIIT), Universiti Teknologi Malaysia, Jalan Sultan Yahya Petra, Kuala Lumpur 54100, Malaysia; yoshihide@utm.my
*   Correspondence: hamizan2816@uitm.edu.my (H.Y.); nurulhuda0340@uitm.edu.my (N.H.A.R.)

**Abstract:** In the 5G system, multiple-input multiple-output (MIMO) antennas for both transmitting and receiving ends are required. However, the design of MIMO antennas at the 5G upper band is challenging due to the mutual coupling issues. Many techniques have been proposed to improve antenna isolation; however, some of the designs have impacts on the antenna performance, especially on the gain and bandwidth reduction, or an increase in the overall size. Thus, a design with a detailed trade-off study must be implemented. This article proposes a new C-shaped parasitic structure around a main circular radiating patch of a MIMO antenna at 16 GHz with enhanced isolation features. The proposed antenna comprises two elements with a separation of 0.32λ edge to edge between radiation parts placed in a linear configuration with an overall dimension of 15 mm × 26 mm. The C-shaped parasitic element was introduced around the main radiating antenna for better isolation. Based on the measurement results, the proposed structure significantly improved the isolation from −23.86 dB to −32.32 dB and increased the bandwidth from 1150 MHz to 1400 MHz. For validation, the envelope correlation coefficient (ECC) and the diversity gain (DG) were also measured as 0.148 dB and 9.89 dB, respectively. Other parameters, such as the radiation pattern, the total average reflection coefficient and the mean effective gain, were also calculated to ensure the validity of the proposed structure. Based on the design work and analysis, the proposed structure was proven to improve the antenna isolation and increase the bandwidth, while maintaining the small overall dimension.

**Keywords:** patch antenna; MIMO; ECC; MEG; DG; surface current distribution and 5G

## 1. Introduction

In order to develop a MIMO antenna system, multiple antenna elements are required for the transmitter and the receiver to achieve a linear increase in the data rate with an increase in the number of antennas. However, it is a challenge to keep multiple antennas within a small and compact space, especially at the user end, due to the mutual coupling effect between the antenna elements [1]. The mutual coupling arises due to the interaction of radiations from closely spaced antennas and the surface currents flowing on the ground plane [2]. These cause field correlation and an increase in mutual coupling and thus degrade the diversity performance of the MIMO antenna system and decrease isolation between adjacent antennas [3].

Various methods have been studied and analysed to suppress the mutual coupling and improve the isolation. Evaluating the antenna surface currents is one of the methods to determine the amount of mutual coupling between the antennas. Some of the MIMO antenna designers introduced parasitic elements [4], orthogonal polarization [5] and decoupling structure [6] to reduce the mutual coupling in MIMO antenna design. However, this method increases the distance between antenna elements. The easiest way to reduce the mutual coupling and improve the isolation between antennas is by adjusting the antenna distance to be more than $\lambda/2$, as mentioned in [7]. Although this method results in high isolation between multiple antennas, this configuration extends the distance between antenna elements; thus, the antenna size is increased. Meanwhile in [8], defected ground structure (DGS) was introduced to reduce the mutual coupling effects. However, the resonant frequency is shifted through this method due to higher coupling caused by the DGS structure. In addition, the DGS itself is a slot antenna, which causes an increase in back lobe radiation.

Studies were conducted in [9–11] to improve isolation and reduce mutual coupling between MIMO elements using metamaterials. The researchers proved that metamaterials can produce good isolation between multiple-element MIMO antennas. The metamaterial works by blocking the transmission of magnetic fields in a near-field environment. At the same time, it improves the far-field radiation and realized gain by increasing the directivity of the antennas [10]. However, the complexity of metamaterial design and structure make this method challenging to implement. Meanwhile, some researchers used stacked configuration to reduce mutual coupling between multiple antennas, as discussed in [12,13]. The multilayer substrate called superstrate was used to reduce the space waves and decrease the mutual coupling between the antenna elements. The multilayer substrate was also able to delay the current path, which can cancel the coupling effect between two closely placed feeding ports [13]. Implementing this stacked configuration was proven to suppress mutual coupling in MIMO systems, but it eventually increases the overall thickness of the antenna design.

In this work, the mutual coupling was reduced by introducing a novel C-shaped parasitic element around the radiating patch structure. The performance of the antenna was investigated through simulation and measurement data. Based on the results, it was proven that the proposed structure can be adopted for 5G MIMO operation due to the high isolation between the elements, improved bandwidth, high gain and small overall dimension.

## 2. Antenna Design and Parasitic Element

Previously, a simple circular patch antenna with a ground plane was designed to demonstrate the capability of the approach in this research, as shown in Figure 1a [14–16]. The antenna was designed to resonate at 16 GHz, as shown in Figure 2. The frequency was chosen to cater for future mobile communication in Malaysia, which is expected to be up to 30 GHz [17]. Then, a C-shaped parasitic element, as shown in Figure 1b, was designed around the main circular patch to improve the antenna bandwidth to support a huge number of users and to cover various applications. The ability of the parasitic element to improve antenna bandwidth and change the radiation resistance is discussed in detail in [18–20]. Meanwhile, apart from enhancing the bandwidth, the parasitic element provides an alternative way for impedance matching [18]. In this design, two parameters were adjusted during optimization. First, the gap between the parasitic element and the center of the radiating patch was adjusted to improve the current distribution. This ensured that the surface current across the strip width and throughout the circular patch was uniform [20]. Second, the size of the parasitic element was adjusted, as the antenna became more inductance when the size was increased [18]. For both optimizations, a high bandwidth was needed.

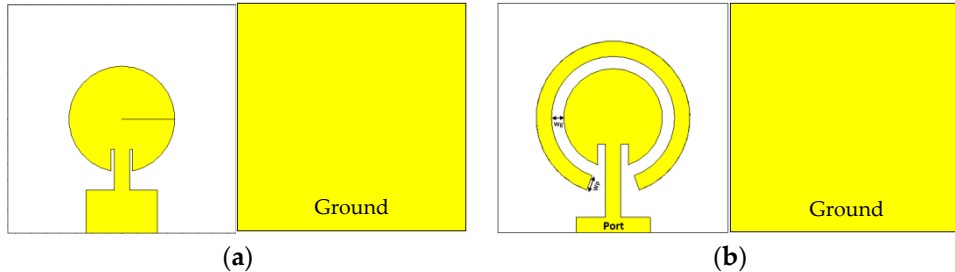

**Figure 1.** Single-element antenna (**a**) without and (**b**) with parasitic elements.

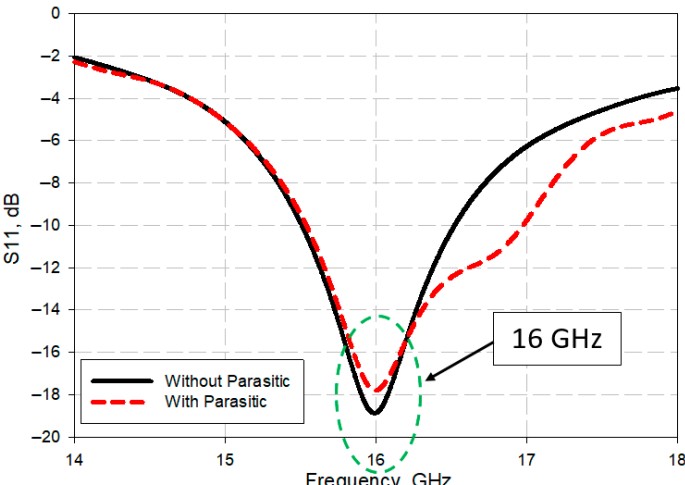

**Figure 2.** S11 performance of both antennas.

Figure 3 shows the behavior of the antenna bandwidth over parasitic gap distance when the parasitic size was changed. The optimization on the parasitic element started with width size, $W_p$, of 1 mm to 3 mm. The distance from the circular patch to the parasitic element, $W_g$, varied from 0.2 mm to 1.0 mm. As shown in Figure 3, the maximum bandwidth obtained was 1459 MHz at $W_g$ = 0.8 mm and $W_p$ = 1 mm. When the $W_p$ was increased to 2 mm, the bandwidth was slightly reduced to 1457 MHz at $W_g$ = 0.4 mm. Then, when the $W_p$ was 3 mm, the maximum bandwidth was only 886 MHz at $W_g$ = 0.6 mm. This scenario indicates a trade-off between the dimension and gap of the parasitic elements and the bandwidth size. Therefore, by considering the size, the most optimum dimensions for the antenna were chosen to be 1 mm and 0.8 mm for $W_p$ and $W_g$ respectively. Table 1 shows the comparison between the antenna performance without and with the parasitic elements. Based on this table, the bandwidth of the antenna increased significantly when a parasitic element was added.

**Table 1.** Single-element antenna performance with and without C-shaped parasitic structure.

| Parameters | Without Parasitic | With Parasitic |
|:---:|:---:|:---:|
| Frequency (GHz) | 16 | 16 |
| Gain (dBi) | 7.98 | 7.69 |
| Reflection coefficient (dB) | −18.74 | −17.98 |
| Efficiency (%) | 80.09 | 81.07 |
| VSWR | 1.2614 | 1.2892 |
| Bandwidth (MHz) | 913 | 1459 |

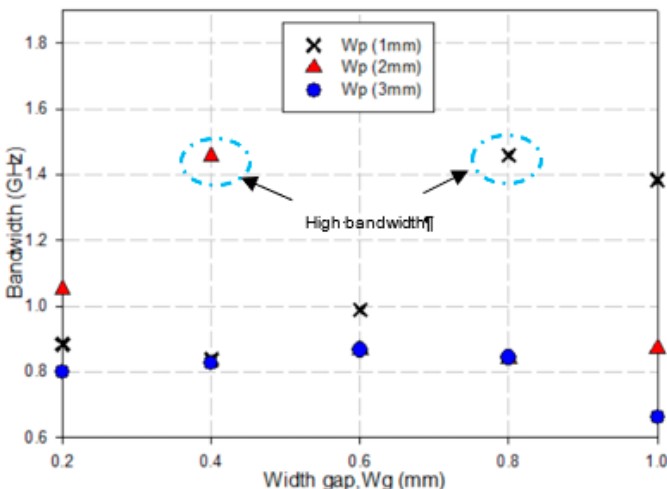

**Figure 3.** Antenna bandwidth for various widths, Wp, and gap distances, Wg.

After optimizing the single element, the design was extended to dual element for multiple-input multiple-output (MIMO) operation, as shown in Figure 4a,b. For comparison, the initially designed single-element antenna without the parasitic element was arranged in a linear 2-element MIMO antenna configuration, as shown by Antenna 1. The isolation between the antennas was monitored when the antennas were placed close to each other. To improve the isolation in dual-antenna configuration, the C-shaped parasitic element, as designed previously, was included in Antenna 2. Besides improving the bandwidth, this parasitic element was used as a wall to block the electromagnetic current from traveling to another antenna element.

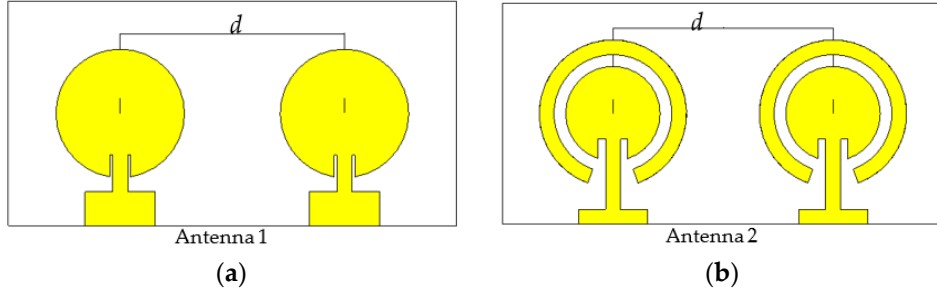

**Figure 4.** Two-element MIMO antenna (**a**) without parasitic elements (Antenna 1) and (**b**) with parasitic elements (Antenna 2).

During optimization, the element spacing, *d* was adjusted from 0.50λ to 0.26λ and the isolation performance was analyzed [21]. For a MIMO antenna to work efficiently, the acceptance/minimum value for isolation must be −15 dB or lower, as mentioned in [22–24]. Figure 5 shows the isolation result concerning the element spacing without and with C-shaped parasitic elements, respectively.

Based on the graph, the antenna without parasitic elements had poor isolation, with a value of −12.33 dB at the desired frequency of 16 GHz. Meanwhile, the isolation value, at −23.86 dB, improved when the C-shaped parasitic element was introduced in Antenna 2, which showed that the mutual coupling was significantly reduced. Furthermore, Antenna 2 showed broader bandwidth as compared to Antenna 1. Therefore, based on the improved isolation value and the bandwidth, a separation of 0.32λ was chosen for the final MIMO antenna design. Moreover, based on Figure 5b, it is clear that by locating the parasitic element around the circular patch, the isolation between multiple antenna elements can be maintained as the spacing increases, as opposed to the scenario in Figure 5a when the parasitic element was not included. It was proven that the parasitic element acts as a perfect isolation tool at a specific frequency regardless of the spacing distance.

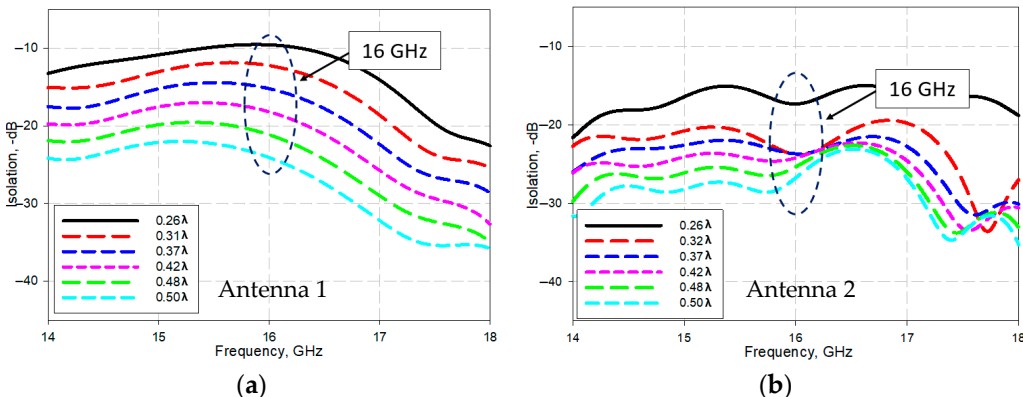

**Figure 5.** Isolation for various spacing distances, *d*, (**a**) without (Antenna 1) and (**b**) with parasitic elements (Antenna 2).

Figure 6 shows the return loss ($S_{11}$) and the transmission loss ($S_{21}$) of Antenna 1 and Antenna 2 with the element spacing, *d*, of 0.32λ. The proposed antenna showed improved $S_{11}$ when both MIMO antenna elements were placed next to each other as compared to Antenna 1. The resonant frequency remained at 16 GHz for Antenna 2 as compared to Antenna 1, where the resonant frequency was slightly shifted to 16.2 GHz. Thus, it was proven that the antenna bandwidth and isolation can be improved by introducing the C-shaped parasitic element without altering the resonant frequency.

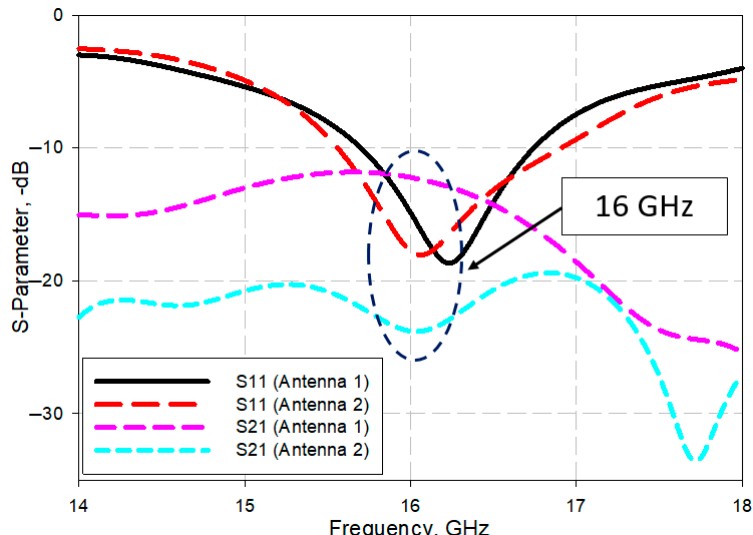

**Figure 6.** Comparison of S-parameters for Antenna 1 and Antenna 2.

Figure 7 shows the final antenna structure with detailed dimensions, as shown in Table 2. The antenna was fabricated on RT-Duroid 5885 with relative permittivity, $\varepsilon_r$, of 2.2, loss tangent of 0.0009 and thickness, *h*, of 1.57 mm. A 50 Ohm feeding line was designed to connect the radiating patch with an electrical source. The main radiating patch was designed on the top substrate layer and a full ground plane was designed on the bottom substrate.

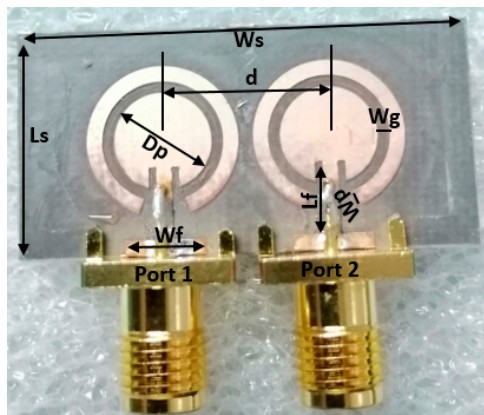

**Figure 7.** Proposed antenna with C-shaped parasitic structure.

**Table 2.** Antenna parameter.

| Parameters | Value (mm) |
|---|---|
| Diameter of patch (Dp) | 3.22 |
| Distance between element (d) | 0.32λ |
| Length of feed (Lf) | 2 |
| Length of substrate (Ls) | 15 |
| Material thickness (Hs) | 1.57 |
| Width of parasitic element (Wp) | 1 |
| Width of feed (Wf) | 4.77 |
| Width parasitic (Wp) | 1 |
| Width of substrate (Ws) | 26 |
| Width gap (Wg) | 0.8 |

The influence of decoupling structure can be observed by visualizing the surface current on the dual-element antennas when the C-shaped parasitic structure was integrated in the design. As shown in Figure 8a, a strong surface current was observed on the patch of Antenna 1. When port 1 was excited, a high mutual coupling could be observed. Meanwhile, the surface current was reduced by introducing a C-shaped parasitic structure around the antennas, as shown in Figure 8b. Thus, it shows that, through the integration of the C-shaped structure, the mutual coupling was reduced. Hence, greater isolation between the antenna was achieved, as was validated further through measurement.

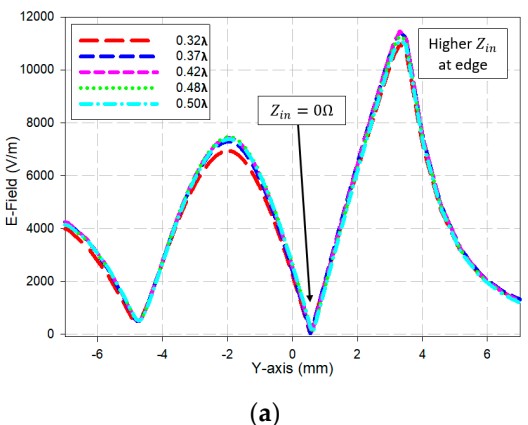

(a)

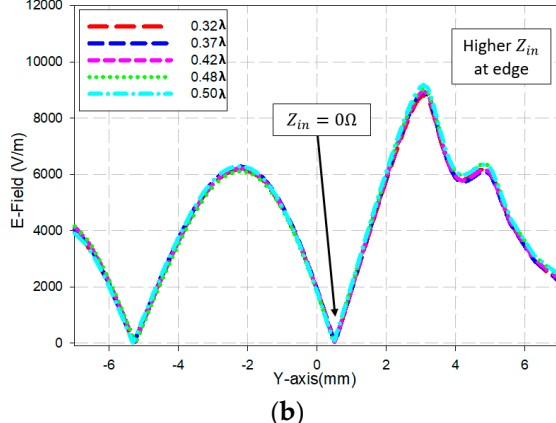

(b)

**Figure 8.** *Cont.*

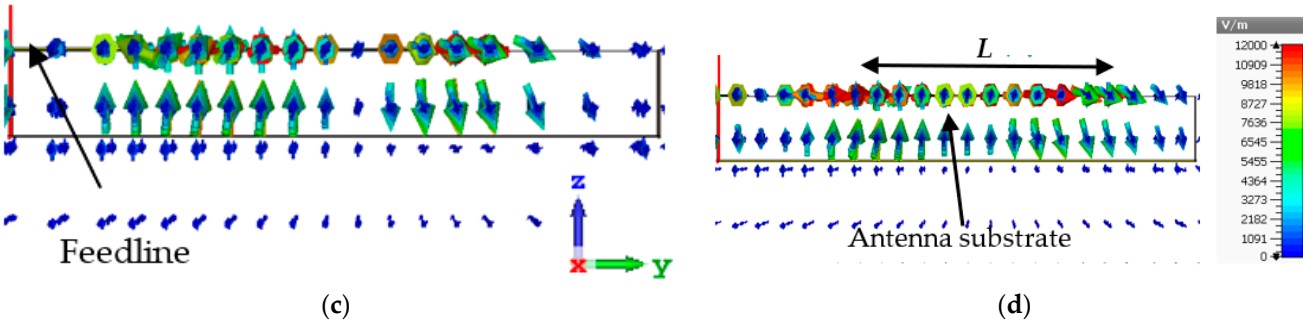

(**c**)                 (**d**)

**Figure 8.** E-field distribution for (**a**) Antenna 1, (**b**) Antenna 2, (**c**) 3D view (without parasitic element) and (**d**) 3D view (with parasitic element).

### 2.1. Observation of Electric Field Intensities along Antenna Edges

As mentioned before, the distance between the two elements affects the antenna isolation when they are located near each other. An electric and magnetic field's intensity graph in the reactive near-field region can be analyzed to validate this condition [25]. Figure 8 shows the electric field (E-field) distribution along non-radiating edges, or length of the antenna, *L*. Theoretically, both graphs show an accurate representation of both conditions, where the E-field intensity is zero at the mid-patch region as the antenna radiating element behaves as a perfect electric conductor at the surface due to the zero impedance ($Z_{in} = 0\ \Omega$). However, it was increasing towards the edge due to the rise in the value of $Z_{in}$. By comparing Figure 8a with Figure 8b, it can be observed that the electric field intensity in the presence of a C-shaped parasitic element was slightly reduced. Based on theoretical current–voltage relation, the observation shows that when a C-shaped parasitic element was added, more energy was radiated to the air instead of trapped inside the substrate or antenna layer, increasing the radiated compared to the absorbed power. This claim can also be verified based on the simulated antenna efficiency, where it shows that the efficiency improved from 65.82% to 80.70% when the parasitic structure was added.

### 2.2. Observation of Surface Currents

Meanwhile, Figure 9 shows the magnetic surface currents along radiating edges, W, of the antenna. The currents were almost uniformly distributed throughout the antenna surface, with an average reading of 15.6 A6 + 9/m and 27.25 A/m for Antenna 1 (Figure 9a) and Antenna 2 (Figure 9b), respectively. It is important to ensure the constant behavior of the current to validate the correlation between E-field intensity and $Z_{in}$. In Antenna 1, without the presence of a parasitic element, as the adjacent distance of the antenna increased, the magnitude of the surface current became low due to the mutual coupling effect between dual-element antennas and the distortion of the shapes of the current distribution. This scenario indicates that the isolation between elements was very poor and had significantly affected the right-hand-side region of the graph, which is the region where both elements were adjacent to each other. When the parasitic element was added to the structure, as shown in Figure 9b, the surface current had uniform distribution at all separation distances. The graph shows that the surface current was unaffected by the distance between the elements. This phenomenon was due to the excellent isolation for Antenna 2, with the presence of a C-shaped parasitic element.

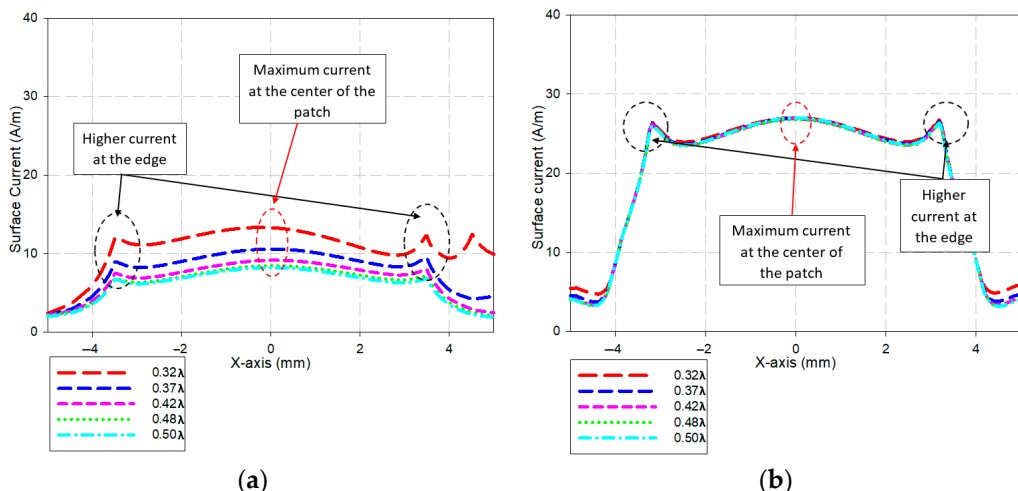

**Figure 9.** Current distribution for (**a**) Antenna 1 and (**b**) Antenna 2.

The influence of the C-shaped parasitic structure can be further observed by visualizing the 3D surface current on the dual-element antennas. As shown in Figure 10a, a strong surface current was observed on the patch of Antenna 1. When port 1 was excited, high mutual coupling could be observed in the adjacent element. Meanwhile, the mutual coupling was reduced by the introduction of C-shaped parasitic structure around the antennas, as shown in Figure 10b, where the adjacent element was isolated when port 1 was excited. Thus, it shows that through the integration of the C-shaped structure, the mutual coupling was reduced. Hence, greater isolation between the antennas was achieved, which was validated further through measurement.

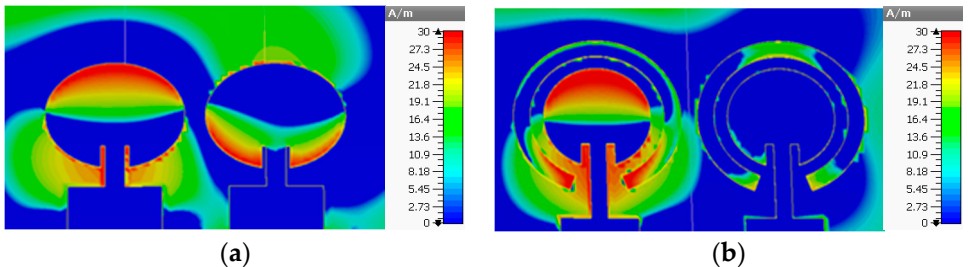

**Figure 10.** Current distribution for (**a**) Antenna 1 and (**b**) Antenna 2.

## 3. Measurement Results and Analysis

The performance of the proposed antenna in linear configuration was measured using the Keysight network analyzer (VNA) and the result is shown in Figure 11.

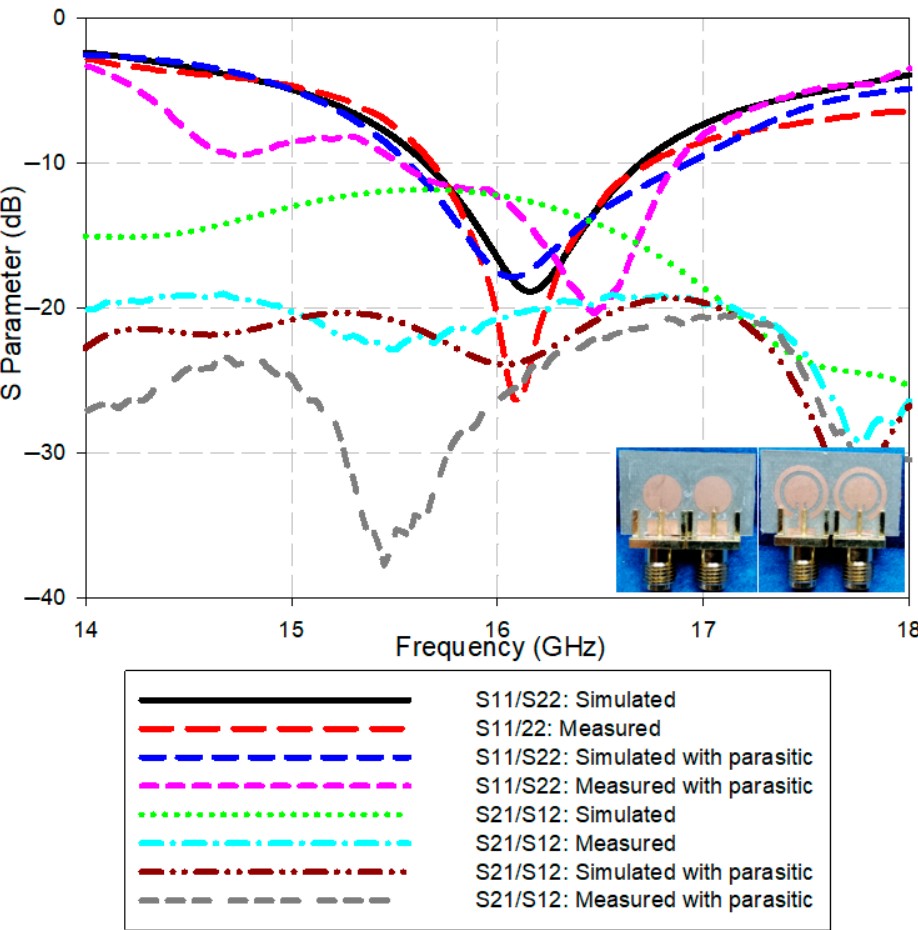

**Figure 11.** Measurement and simulation results of S-parameter for Antenna 1 and Antenna 2.

### 3.1. Return Loss and Isolation

The graph in Figure 11 shows that the measured $S_{11}$ is slightly shifted from the simulated data. The discrepancies are due to the alignment at the SMA connector and fabricated tolerance. However, the difference is very small; thus, the result is acceptable because the bandwidth covers the desired frequency band.

As observed in Figure 10b, the C-shaped parasitic element can block the surface waves inside the antenna substrate and guide them in another direction by creating an indirect signal with the additional coupling path that opposes the signal going directly from an element to another element. Therefore, with the measured data in Figure 11, the theory is proven through the reduction in mutual coupling, resulting in an isolation of −32.32 dB and reflection coefficient of −15.17 dB over 15.56 GHz to 17.00 GHz bandwidth.

### 3.2. Radiation Pattern and Antenna Gain

To observe the performance of the parasitic element structure on the radiation pattern of the antenna, the simulated and measured E-plane and H-plane data are shown in Figure 12. It is observed that the radiation pattern is dominant at 0° angle and good agreement is obtained between the measured and simulated radiation patterns. The proposed antenna is able to produce a gain of 6.21 dB (Antenna 1) and 6.43 dB (Antenna 2), which is more desirable. Although the measured gain is less than the simulated result (7.98 dB Antenna 1 and 7.69 dB Antenna 2), the proposed antennas meet the research requirement for design, as the MIMO antenna shows a gain of more than 5 dBi. Meanwhile, the measurement result for the antenna efficiency is 53.58% (Antenna 1) and 59.17% (Antenna 2). The efficiency is directly proportional to the value of gain [26]. The lower efficiency in measurement reduces the total gain measurement.

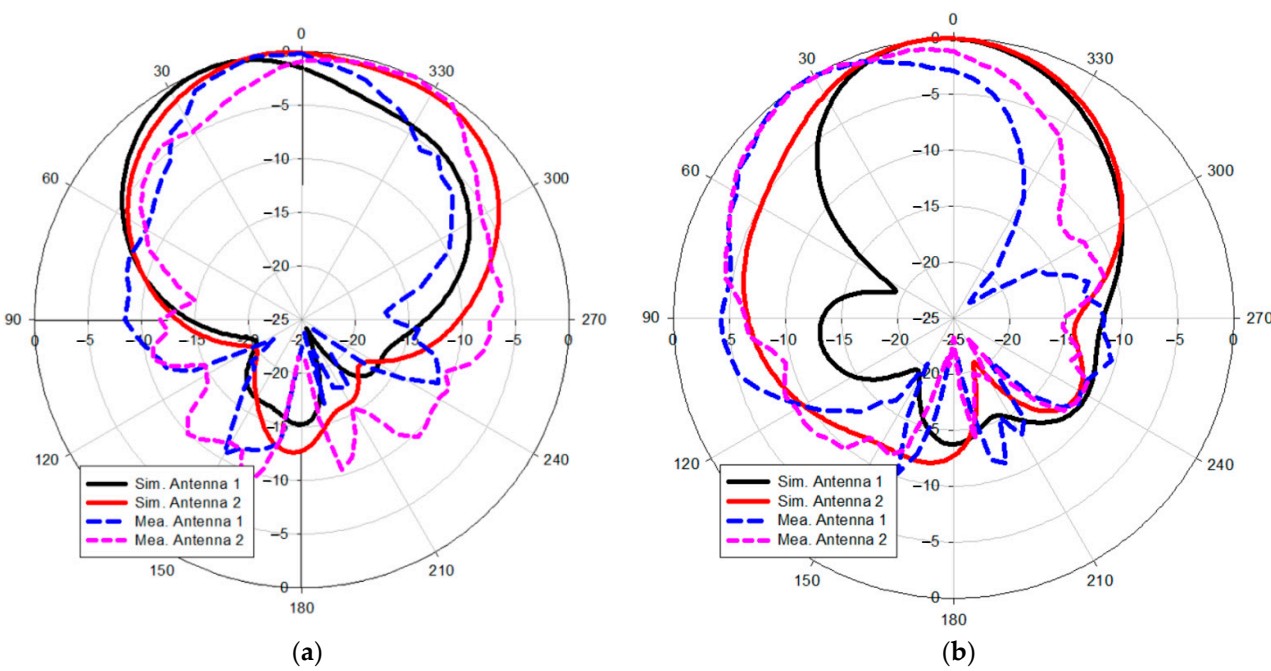

**Figure 12.** Measured and simulated radiation pattern for MIMO antenna: (**a**) E-plane and (**b**) H-plane.

### 3.3. MIMO Performance Analysis

In this section, MIMO parameters, such as the total active reflection coefficient (TARC), envelope correlation coefficient (ECC), diversity gain (DG) and mean effective gain (MEG) are evaluated to validate the MIMO characteristics for multipath propagation [27].

The total active reflection coefficient (TARC) is defined as the ratio of the square root of the total reflected power divided by the square root of the total incident power in a multiport antenna system [28]. TARC is a method of manipulating all S-parameters for all MIMO ports and displaying them on a single curve, representing the effects of the feeding phase on the antenna port [29]. The value was obtained randomly with a phase swept between 0° and 180° to create the TARC curve [30]. The measured TARC of the proposed antenna design with parasitic elements is shown in Figure 13. The result indicates that the TARC of the proposed antenna covers the desired band. For a two-port MIMO antenna system, TARC can be evaluated using (1) [30].

$$\tau = \frac{\sqrt{((|S_{11} + S_{12}e^{j\theta}|^2 + (|S_{21} + S_{22}e^{j\theta}|^2))}}{\sqrt{2}} \tag{1}$$

The correlation coefficient (ρ) measures how much the communication channels are isolated or correlated between multiple MIMO antennas. This parameter considers the antenna radiation pattern when two or more antennas are working simultaneously. Meanwhile, the square of the correlation coefficient value is known as the envelope correlation coefficient (ECC) [31]. The envelope correlation coefficient can be calculated as mentioned in [32]. However, it is only valid for a uniform multipath environment, such as an isotropic environment. This equation can be simplified using the S parameter [33], as shown by (2). Ideally, the ECC value should be zero [34]. However, in [29] the maximum value was set to 0.5, an acceptable value for wireless systems. As shown in Figure 14, the ECC values for antenna without and with parasitic elements are 0.1408 and 0.1482, respectively.

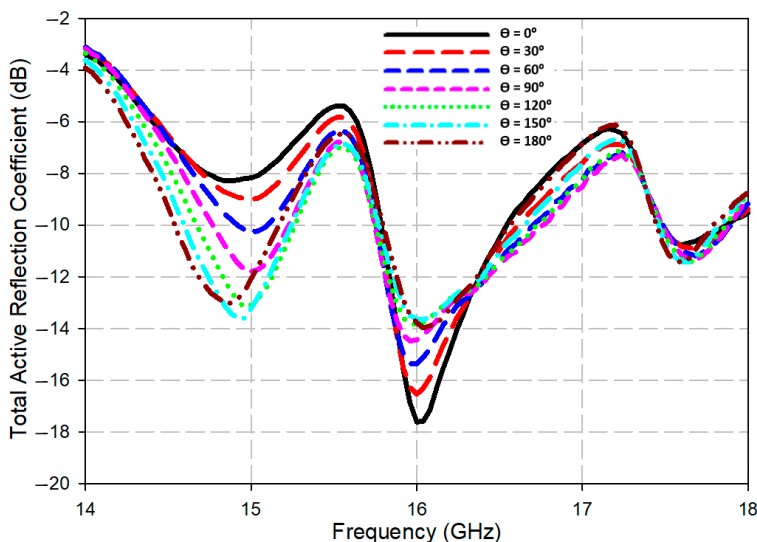

**Figure 13.** Measurement of total average reflection coefficient (TARC).

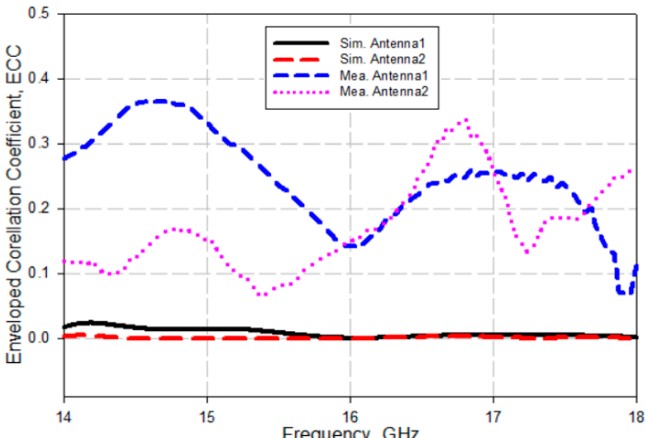

**Figure 14.** Measurement and simulation of the enveloped correlation coefficient (ECC).

$$Pe = \left| \frac{|S*_{11} S_{12} + S*_{21} S_{22}|}{\left| \left(1 - |S_{11}|^2 - |S_{21}|^2\right)\left(1 - |S_{22}|^2 - |S_{12}|^2\right) \right| \frac{1}{2}} \right| \tag{2}$$

Diversity gain (DG) is a measure of the effect of diversity on communication systems [29]. The diversity occurs when a transmitter receives multiple signals from different source channels in MIMO systems. The parameter must be higher (uncorrelated) for better signal reception. The more an antenna is used in a MIMO system, the more the combined power is received in a diversity system. Diversity gain and correlation coefficient are interrelated. The higher the diversity gain means the lower the correlation coefficient. The DG can be calculated by using the equation as given in [35]. Figure 15 shows the comparison between the simulated and measured DG of the proposed MIMO antennas to verify this result. The measured values for Antenna 1 and Antenna 2 are 9.9024 dB and 9.8918 dB, respectively. This observation indicates that the DG of the proposed antenna is nearly at the maximum value of 10 dB, as mentioned in [36]. This is a productive value for the antenna because the lower the correlation coefficient, the higher the diversity gain.

$$DG = 10e_p \tag{3}$$

$$e_p = \sqrt{1 - |\rho|^2} \tag{4}$$

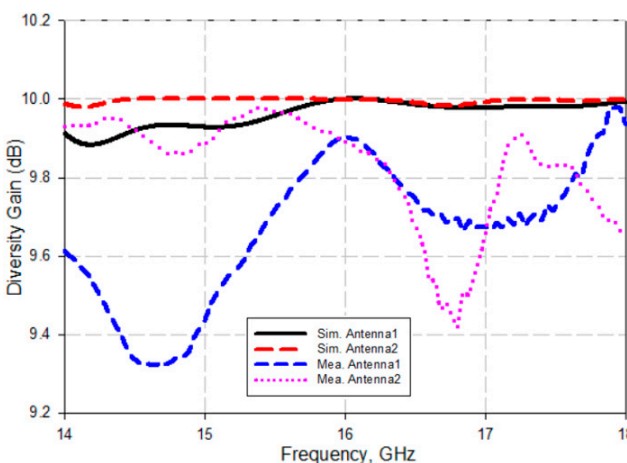

**Figure 15.** Measurement and simulation of the diversity gain (DG).

Mean effective gain (MEG) is defined as a measure of the gain performance in a predefined wireless environment where the effect of the environment is taken into account [37]. The MEG is an important parameter to determine the antenna performance in the real environment. The easiest way to determine the MEG is by using a practical calculated method, as described in [38]. The technique is also discussed in [39]. The practicality of the environment was replaced by using a 3D radiation pattern and the proposed statistical model that can achieve MEG numerically by solving a mathematical expression that combines the two quantities. The MEG can be determined using a measured/simulated gain pattern in an ideal environment with this concept. Meanwhile, in [27] the MEG was calculated from the S-parameter (simulation/measurement) results using (5), where M is the total number of antennas [40]. The power ratio (k), which is the difference in the magnitude of MEG, was calculated using (6). Figure 16 shows that the measured MEG at selected frequency is 0.3514 dB and 0.1482 dB, respectively, for designs with and without parasitic elements. Meanwhile, the k (power ratio) value is almost 0 dB, which means that there is no significant difference between the average power received [27] at the proposed MIMO antenna design.

$$MEG_i = 0.5\eta_{i,rad} = 0.5\left[\sum_{j=1}^{M} |S_{ij}|^2\right] \tag{5}$$

$$k = |MEG_1 - MEG_2| < 3\text{dB} \tag{6}$$

Table 3 shows the comparison between the existing antenna design with the proposed MIMO antenna in terms of bandwidth, isolation, ECC and diversity gain. Based on the observation of previous antenna designs, some designers have ignored the diversity gain and enveloped correlation coefficient parameters that are very important to measure the MIMO antenna total performance. Meanwhile, the integration concept with a C-shaped parasitic element is proven to be able to improve the isolation and the antenna bandwidth. Additionally, with a smaller size than other designs, this antenna is suitable for future mobile applications.

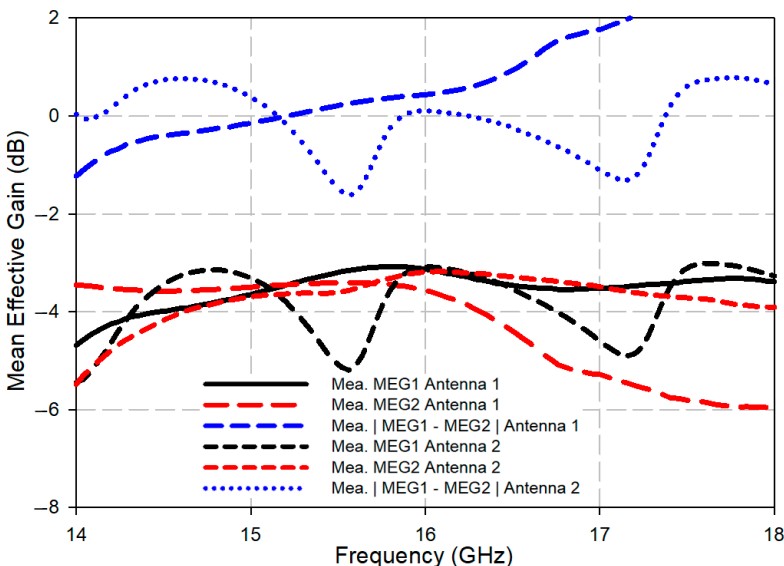

**Figure 16.** Measurement of the mean effective gain (MEG).

**Table 3.** Comparison between the proposed and existing MIMO antennas.

| Cite | Size (mm$^2$) | Bandwidth (GHz) | Isolation (dB) | TARC (dB) | ECC (dB) | Diversity Gain (dB) | MEG (dB) |
|---|---|---|---|---|---|---|---|
| [41] | 2.89λ × 1.03λ | 26 to 31 | >−21 | - | 0.15 | - | - |
| [42] | 1.77λ × 2.42λ | 28 to 28.5 | >−25 | - | - | - | - |
| [43] | 5.6λ × 2.96λ | 15.6 to 17.1 | >−20 | - | - | - | - |
| [44] | 2.5λ × 2.5λ | 3.1 to 17.3 | >−15 | - | 0.1 | - | - |
| [45] | 1.13λ × 1.13λ | 3 to 30 | >−20 | - | - | - | - |
| [46] | 0.9λ × 0.5λ | 3 to 12 | >−20 | - | 0.2 | - | - |
| Proposed | 0.8λ × 1.33λ | 15.5 to 17 | >−30 | Yes | 0.14 | 9.89 | 0.351 |

## 4. Conclusions

Detailed analysis on the ability of the new antenna structure with a C-shaped parasitic element to suppress the mutual coupling between the MIMO antenna elements was successfully demonstrated. The antenna performance was validated through fabrication and measurement, and excellent results were obtained. The proposed antenna is well matched at the resonant frequency of 16 GHz, and it is suitable for 5G mobile applications. The C-shaped element was proven to block the surface current due to the high value of isolation between the MIMO elements. The result shows that the mutual coupling was improved by 8.58 dB, from −23.74 dB to −32.32 dB when the C-shaped parasitic element was added. The improvement is excellent for a typical MIMO system that requires isolation of more than −15 dB. The relevant MIMO parameters were measured (ECC < 0.1482), (DG > 9.8918) and (MEG > 0.3514) in the operating band. The radiation pattern of the MIMO antenna was also measured in an indoor OTA500 chamber facility. Although the gains were slightly less than the simulated values, the obtained values are acceptable with 6.21 dB for Antenna 1 and 6.43 dB for Antenna 2, showing that the parasitic element improved the radiation performance.

**Author Contributions:** Conceptualization, H.Y. and H.J.; methodology, I.K.C.L. and M.H.J.; formal analysis, N.H.A.R. and Y.Y.; writing—review and editing, N.H.A.R.; supervision, N.H.A.R., M.H.J. and M.A.A.; funding acquisition, F.N.M.R. and M.A.A. All authors have read and agreed to the published version of the manuscript.

**Funding:** This research was funded by Universiti Teknologi MARA through the Young Talent Researcher Grant (No.: 600-RMC/YTR/5/3 (012/2020).

**Acknowledgments:** This work was supported by Universiti Teknologi MARA through the Young Talent Researcher Grant (No.: 600-RMC/YTR/5/3 (012/2020). Authors would also like to thank all researchers of Antenna Research Centre, College of Electrical Engineering, Universiti Teknologi MARA, Shah Alam, Selangor, Malaysia for supporting this project.

**Conflicts of Interest:** The authors declare that they have no known competing financial interests or personal relationships that could have appeared to influence the work reported in this paper. The funders had no role in the design of the study; in the collection, analyses, or interpretation of data; in the writing of the manuscript, or in the decision to publish the results.

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
