# Peer review of "Development of C-Shaped Parasitic MIMO Antennas for Mutual Coupling Reduction"

_electronics, doi:10.3390/electronics10192431_

Round 1
Reviewer 1 Report
This paper presented a circular antenna with a C shape parasitic structure. The antenna presented improved bandwidth and isolation compared with original circular antenna. The simulation and measurement results supports the authors arguments, however, there are a few issues that need to be addressed before it can be published:
- The author did not describe the antenna configuration clearly. Is it a circular dipole antenna (no gnd plane under patch) or circular patch antenna (with gnd plane under patch)?
- In table 1, the author claimed that the resonance frequency is at 16GHz for single antenna. The author should present the S11 for both single antenna. Otherwise, how would the reader believe that antenna1 resonance frequency actually moved by 0.2 GHz due to coupling?
- Figure 7 and 8 are confusing. The author should be more specific about where are these fields got measured. (what is non-radiating edges, L of the antenna?) A figure is preferred to illustrate the case.
- At what frequency are the fields measured in Fig7 and 8 and 9? Are they both at 16 GHz? If the Antenna1 resonance at 16.2 GHz, would it be better if we plot the antenna1 field at 16.2GHz?
- Table3, due to different substrate and different operation frequency, and different number of antennas in the array, it is not fair to directly compare the physical sizes between different works.
Author Response
Dear reviewer, thank you very much for your valuable comments and suggestions. The authors have revised the original manuscript according to the comments, and have made significant modifications to the content to improve the quality of the overall manuscript. The changes made are shown in RED font in the manuscript.

Reviewer 2 Report
In this work, authors design manufacture, and measure a MIMO antenna system for 5G (16GHz). They improve upon their baseline circular radiating patch (antenna element) with a parasitic C-shaped structure. The final design (and implementation) is a two-element antenna with 32dB of isolation and 1400MHz bandwidth. Along with typical antenna parameters as radiation pattern, total average reflection coefficient, and the mean effective gain, authors also present the MIMO antenna’s envelope correlation coefficient (ECC) and the diversity gain (DG).
Comments:
- Page numbering is wrong (top right-hand side of the page)
- I strongly advise improving the quality of the Figures in the paper. They start to blur with the slightest zoom attempt.
- Figure 9 lacks a colormap, dimensions, or any quantitative information.
- Add to the figure 4 caption that it is with(antenna1) /without(antenna2) parasitic element for clarity.
- At the introduction section, more methods should be added in detail (e.g. metamaterials) with the appropriate references.
- As a general comment, I don’t think section 2 (after line 158) is necessary/ or adding any information since the isolation improvement has been proven adequately beforehand in the same section.
- Should the authors select to keep this part of section 2, I suggest a new sub-section be formed at line 158 since the following text attacks the problem from a different perspective.
- Also, further details should be provided on the radiating and non-radiating edges, as well as on figures 7 & 8 regarding the Y-axis (mm) notation. e.g. the scale is approximately -6.5 mm to 6.5 mm. This scale should be identified clearly on the antenna or clearly described on which exactly side it is referring to. Since this part of the paper follows the work of [22] (per the manuscript references), I need to comment that in [22] the x,y,z axes hence the radiating and non-radiating edgers are clear to the reader. In the present work, however, this is not the case. This should definitely be corrected by clearly explaining the approach and its application in the present paper.
- At the conclusion section, I strongly disagree with expressing the improvement as a percentage. The improvement should be simply expressed in dB.
- A short discussion on the performance of the antenna with respect to the relevant MIMO parameters should be included in the conclusion section (also in section XX) to highlight the overall MIMO performance of the antenna. e.g., minimum acceptable/or target values; are the authors satisfied with every parameter? Do they need to/would they like to improve a particular parameter? What are their next steps in order to do so?
Author Response
Dear reviewer, thank you very much for your valuable comments and suggestions. The authors have revised the original manuscript according to the comments and have made significant modifications to the content to improve the quality of the overall manuscript. The changes made are shown in RED font in the manuscript.

Round 2
Reviewer 1 Report
The authors answered my questions properly. The paper can be published
Reviewer 2 Report
The authors have improved their work by adding all the missing information on the revised paper according to the reviewer's comments.